# Cyclic Voltammetric-Paper-Based Genosensor for Detection of the Target DNA of Zika Virus

**DOI:** 10.3390/mi13122037

**Published:** 2022-11-22

**Authors:** Anirudh Bishoyi, Md. Anish Alam, Mohd. Rahil Hasan, Manika Khanuja, Roberto Pilloton, Jagriti Narang

**Affiliations:** 1Department of Biotechnology & Microbiology, National College (Autonomous), Tiruchirapalli 620001, India; 2Department of Biotechnology, School of Chemical and Life Sciences, Jamia Hamdard, Hamdard Nagar, New Delhi 110062, India; 3Centre for Nanoscience and Nanotechnology, Jamia Millia Islamia, New Delhi 110025, India; 4Institute of Crystallography, National Research Council (IC-CNR), 00118 Rome, Italy

**Keywords:** Zika virus, genosensor, paper electrodes, cross-reactivity, synthetic serum

## Abstract

**Highlights:**

**Abstract:**

Zika virus (ZIKV), a positive-sense single-stranded RNA virus, has been declared as the cause of a ‘worldwide public health emergency’ by the WHO since the year 2016. In cases of acute infections, it has been found to cause Guillain–Barre syndrome and microcephaly. Considering the tropical occurrence of the infections, and the absence of any proper treatments, accurate and timely diagnosis is the only way to control this infectious disease. Currently, there are many diagnostic methods under investigation by the scientific community, but they have some major limitations, such as high cost, low specificity, and poor sensitivity. To overcome these limitations, we have presented a low-cost, simple-to-operate, and portable diagnosis system for its detection by utilizing silver nanoparticles. silver nanoparticles were synthesized via chemical methods and characterization was confirmed by UV/TEM and XRD. The paper platform was synthesized using a graphene-based conductive ink, methylene blue as the redox indicator, and a portable potentiostat to perform the cyclic voltammetry to ensure true point-of-care availability for patients in remote areas.

## 1. Introduction

A mosquito-borne flavivirus, ZIKV, has been the cause of multiple public health concerns and a long-running epidemic since 1947. Previously limited to occasional instances in Africa and Asia, the virus’ outbreak in 2015 in Brazil suggested an immediate spread throughout the US. The WHO declared it a worldwide public health emergency in 2016 [1]. ZIKV has recently emerged as a major worldwide threat because it causes microcephaly in infants born to infected mothers. Epidemiological monitoring of infection has been hampered by the lack of reliable tests capable of distinguishing between Zika and other Flavivirus infections, especially the dengue virus. Various Zika-based biosensors have been developed, which offer advantageous features such as low detection limit, linear range, and stability, but have some limitations such as the use of the bulky three-electrode setup and high reagent/sample volume, which restrict their application as POC-based devices. ZIKV, like other Flaviviruses, is primarily an enveloped virus with a single-stranded RNA (ssRNA) genome of nearly 11Kb +ve polarity-based gene encoding a single polyprotein, bound by an icosahedral enclosure comprising 180 copies within every envelope (E) glycoprotein with about 500aa (amino acids), and approx. 75aa membrane protein and approx. 165aa precursor-membrane ‘prM’ protein [2,3] Symptoms include moderate influenza, neurological symptoms, and subclinical presentations in a child born to an infected mother, and Guillain–Barre syndrome (GBS) in adults [4]. There is currently no cure for ZIKV. The ZIKV may be detected quickly and easily, lowering the risk of infection and improving its management. Traditional ZIKV testing methods, such as virus isolation, identification of ZIKV-specific antibodies, ELISA, and PCR, have some drawbacks. There is a requirement for an immediate and uncomplicated technique for the diagnosis of ZIKV [5,6,7,8,9,10]. Scientists are very interested in developing an effortless and portable approach for diagnosing ZIKV. Screen-printed electrodes (SPEs) can be fabricated from a variety of materials, including plastic, paper, and tattoos, and they provide a lot of flexibility in terms of size, shape, dimensionality, and customizing options. SPEs could be especially useful in the development of new approaches and research. Screen printing is distinguished by its ease of production and application, as it does not necessitate the usage of cleanrooms and can be performed with low-cost components and even homemade setups. Conductive inks, an electrode-patterned mesh screen, a squeegee, and an oven are all that are needed for the screen-printing process [11].

Nanobioconjugates are nanomaterials that have been combined with biomolecules to form hybrid nano (bio) materials. Hundreds of functional biomolecules and tags immobilized on nanomaterials(nanobioconjugates) have been proven to boost signal enhancement in a variety of biosensors. Such hybrid systems outperform their individual components, with each component contributing a unique trait or function to the hybrid that the other does not [12]. The conjugation of AgNPs with single-stranded DNA (ssDNA) provides a simple way to create nanobioconjugates with exceptional capabilities for signal extension, improved sensitivity, and reduced LOD in bioassays [13]. Alves et al. [14] developed a genosensor based on the graphite carbon electrode. A DNA probe was immobilized covalently onto the sensing interface and hybridization was detected by the differential current response. The major advantage of the developed sensor was that it was applied in an experimental sample and interference with other flaviviruses was also studied. Another Zika-based genosensor was developed on SPEs by Cajigas et al. [12]. SPEs were modified with gold nano-architecture and Ru was used as a labeling agent. The sensor was also applied in serum samples, but the SPEs are very expensive and corrode easily, which limits their application in POC-based devices. Faria et al. [15] developed a disposable electrode, which offers many advantageous features such as being economical, having a low reagent volume, and no bulky electrode set up. The developed biosensor has some limitations, such as higher response time (1.5 h), and the sensor was not tested to an experimental sample. Paper-based carbon conductive SPEs modified with AgNPs were utilized as a platform for immobilization of the oligonucleotide probe of ZIKV. This substance is simple and economical, has a high reaction yield, an extensive surface area, and the ability to be functionalized, and is biocompatible. Thus, taking into account things such as enhanced performance, this effort represents a novel platform for detecting ZIKV, including the potential to fill the gap in the lack of approaches that sense the target at very low concentrations, while also delivering ZIKV analysis in the sample with stability, portability, and quick analysis. Therefore, PBG is a suitable substitute for easy, stable, low-cost, and quick identification. As a consequence of the high specificity of DNA, electrochemical DNA sensing approaches have received a lot of attention. The precision and sensitivity of DNA-based electrochemical biosensors has improved due to the use of nanomaterials for signal improvement. Specifically, the working electrode of PBG was modified with silver nanoparticles on account of its high electrical and thermal conductivity. Furthermore, the biosensor specific to the target TDNA of ZIKV was immobilized on the surface-modified working electrode of PBG and an electrochemical investigation was executed to verify the respective hybridization.

The current study aimed to develop an electrochemical paper-based device for the detection of ZIKA virus target DNA, which has many advantages such as low sample/reagent volume, ease of preparation, and more stable printing on paper versus plastic substrate, and the sensor was also tested in artificial serum samples to ensure sensor applicability in complex matrices. The use of a cutting-edge portable potentiostat limits the device’s LOD, but this is mitigated by the fact that it can be replicated in any remote corner of the world, effectively making the device truly capable of providing POC diagnosis, which is critical taking the tropical presence of the disease under consideration.

The analytical performance of AgNP/PBG-based biosensor for the selective recognition of ZIKV TDNA was confirmed in this study, establishing the created sensor as a possible candidate for precise and sensitive Zika virus diagnosis.

## 2. Methods

### 2.1. Chemicals, Reagents, Apparatus

Methylene blue was purchased from Sigma Aldrich, India, and all the other chemicals were of AR grade: silver nitrate (Qualigens) and sodium borohydride (GLR Innovations). ZIKA target was dissolved in sterile water (100 µM). For the making of synthetic serum: NaCl, CaCl_2_, KCl, MgSO_4_, NaHCO_3_, Na_2_HPO_4_, and NaH_2_PO_4_ were used. A 23-base oligonucleotide probe DNA(PDNA) was obtained from Merck (Rahway, NJ, USA) for capturing the 23-base target DNA (TDNA), which it was complimentary to. The used sequence of ZIKV target DNA and probe DNA was acquired from the work of Zhang et al. [16].

Probe Sequence: AGCCATGACCGACACCACACCGT

Target Sequence: TCGGTACTGGCTGTGGTGTGGCA

The yellow fever oligonucleotide, used for testing the specificity of the device, was also obtained from Merck (Rahway, NJ, USA).

Target Sequence: CGATTAACTCCACATAACCAGACG

For the preparation of silver NPs: chloroauric acid (John Baker Inc.), trisodium citrate (LOBA), NaBH4 (GRL Innovations), CTAB (HIMEDIA), and ascorbic acid (SRL) were used. All other chemicals are purchased from LOBA (Colaba, Mumbai, India).

The electrochemical measurements including cyclic voltammetry (CV) were performed on Metrohm Dropsens (Asturias, Spain) (µStat-I 400s). Photometric analysis of absorbance was performed using UV-Visdouble beam spectrophotometer (HALO DB-20R) (Dynamica Scientific, Livingston, United Kingdom). The microstructural characterizations were executed via using X-ray diffraction (XRD) (Smart Lab guidance, Rigaku, Tokyo, Japan) to check the phase and crystallinity. AgNP morphology was inspected via TEM on Talos L120C (Thermo Fisher Scientific, Waltham, MA, USA).

### 2.2. Synthesis of Silver NPs

Silver NPs were synthesized by the chemical reductant method. A total of 10 mL of freshly prepared 1mM AgNO_3_ solution was added in a dropwise fashion (1 drop/second) to 30 mL of 2 mM ice-cold NaBH_4_ solution, while continuously stirring the solution, to form a bright yellow solution when all the silver nitrate was added; this bright yellow solution contains the AgNPs.
AgNO_3_ + NaBH_4_→Ag + ^1^/_2_H_2_ + ^1^/_2_B_2_H_6_ + NaNO_3_

The bright yellow solution containing the Ag nanoparticles was analyzed by photometric analysis and showed peak absorbance at 387.5 nm using a HALO DB-20R UV-Vis beam spectrophotometer.

### 2.3. Preparation of Synthetic Serum

For the preparation of synthetic serum, 6.8 g of NaCl, 0.2 g CaCl_2_, 0.4 g KCl, 0.1 g MgSO_4_, 2.2 g NaHCO_3_, 0.126 g Na_2_HPO_4_, and 0.026 g of NaH_2_PO_4_ were dissolved in distilled water, and the pH was maintained at 7.4.

### 2.4. Development of PBGs

A silk screen with a laser-cut patterned solid skin adhered to it, according to predetermined dimensions for a two-electrode system, was used for hand printing. Carbon conductive ink was pressed onto cellulose papers through the open regions of the overhead specified screen using a squeegee. The dimensions of the electrode were pre-fixed and framed on the silk screen, which was further used as a stencil for the preparation of the electrodes. The printed electrodes were a two-electrode system: a counter electrode (CE) and a working electrode (WE). This led to the construction of the PBGs.

### 2.5. Deposition of the Silver Nanoparticles and Immobilization on the PBG

The synthesized silver nanoparticles were drop-deposited (20 µL) on the circular working area of the paper-based biosensor. The paper-based biosensor was then dried on a hot plate at 60 °C. After modification with silver nanoparticles, the working area was immobilized with the consensus biosensor of the probe. For this, 20 µL of the probe DNA was dropped over the working area. This biosensor-modified electrode was further used for the recognition of the target DNA of Zika.

### 2.6. Stages for Electrochemical Detection

In order to make a functioning and selective biosensor for detecting Zika virus cDNA, the probe DNA needs to be deposited onto the working electrode with the help of the nanoparticles. For this, cyclic voltammetry (CV) values of bare electrodes with no deposition were analyzed. Next, the AgNPs were deposited onto the paper-based biosensor and dried over night after which each voltammetry was repeated as before; for the next step, the PDNA was deposited on a paper-based biosensor containing dried AgNPs, and CV values were recorded at each fabrication stage. For the last phase of the biosensor design, the analyte molecule, the TDNA, was deposited onto electrodes containing both AgNPs and PDNA; CV was performed and values were recorded.

### 2.7. Optimization of Physicochemical Parameters

The detection capacity of the genosensor was optimized by observing the alterations in the voltammograms caused by changes to the different detection parameters. This was achieved by manipulating the required parameters in the TDNA sample. Concentrations of 0.1, 1, 10, and100 µM were made, the TDNA was incubated at different temperatures (15, 25, 30, and 45 °C), and the time given for hybridization by the potentiostat (tcond) was altered.

### 2.8. Binding of the Analyte on AgNPs/Probe/Target/PBGs

The different concentrations of the biosensor (0.1, 1, 10, and100 µM) were detected by dropping a mixture of target DNA and MB. The CV measurements were performed so that the hybridization between the biosensor and target DNA could be confirmed. TDNA was drop-deposited over PDNA/AgNPs/PBGs with different concentrations of TDNA.

### 2.9. Procedure for Experimental Sample Analysis, Repeatability, and Stability Analysis

The capability of the genosensor to perform in experimental samples was checked by adding a known concentration of the target DNA in the synthetic serum. This solution along with the hybridization indicator was added over a probe biosensor-modified paper-based electrode system. Electrochemical assessments were performed to check the results.

### 2.10. Principle behind Sensing

The proposed PBG works on the principle that the hybridization of DNA into the double-helical structure causes a lowering of the peak current value of the CV curve, as demonstrated by Pan et al. [17]. This can be attributed to the molecular interaction of MB with ssDNA and dsDNA, while with ssDNA it can only have electrostatic interactions; however, when the DNA is present in its double-stranded helical form, it can also intercalate between the structure in addition to the electrostatic interactions. Therefore, in ssDNA, MB has a significantly lower binding constant, thus allowing for a better passage of current. This results in a greater negative formal potential with dsDNA than with ssDNA.

## 3. Results

The fabrication method of the electrode involves the deposition of AgNPs onto the working surface of the paper electrode. The AgNPs were drop-deposited onto the surface of the electrode. After the drop-deposition of the AgNPs, the DNA probe was also put onto the working surface of the paper electrodes. The AgNPs provided a biocompatible environment to the biological identification component and accelerated the electron transfer.

### 3.1. Morphological Characterization of Silver Nanoparticles

The XRD patterns of silver nanoparticles are shown in Figure 1a. The XRD pattern revealed that it has a face-centered cubic crystal structure. According to the obtained XRD result, the 111, 200, 220, and 311 crystallographic planes of silver nanoparticles were responsible for the 2thata values of 38.15°, 44.06°, 66.20°, and 76.61° peaks, respectively. The XRD pattern demonstrated that the Ag-NPs synthesized in this work were crystalline in nature. The formation of Ag-NPs was also confirmed by measuring the SPR over the wavelength range of 300–800 nm using UV-Vis spectroscopy. According to the reported data, the spherical Ag-NPs contribute to the absorption bands at approximately 400–450 nm in the UV-visible spectra. The UV-visible absorption spectrum revealed that the wide SPR band had a single peak at 445 nm, which was located in the broad SPR band. This peak confirmed the presence of a uniform distribution of Ag-NPs in the synthesized sample as shown in Figure 1b. TEM images (Figure 1c) depicted that the silver nanoparticles are spherical in shape and 30 nm in size. Figure 1d shows the average particle size of the nanoparticles to be around 25nm.

### 3.2. Electrochemical Properties of DNA Probe/AgNPs/PBGs

The electrochemical characterization of the DNA probe/AgNP/PBG-modified electrodes was performed by employing the electrochemical CV technique. Figure 2 shows the differential current response at different stages of the electrode. As the bare electrode showed a smaller peak current response (0.26 µA), which is attributed to the lower electron transfer kinetics. Upon deposition of the AgNPs onto the working surface, there was a significant two-fold increase in current response (1.59 µA), which was due to the fast electron transfer kinetics provided by silver nanoparticles. After immobilization of the biological recognition element (DNA probe) onto the working surface, the current (1.07 µA) was drastically decreased due to the nonconductive nature of DNA. After the introduction of target DNA, the current response (0.81 µA) was further decreased due to the well-known MB principle. MB was intercalated between the bases, which significantly decreased the current response.

### 3.3. Effect of Different Target DNA Concentrations on the DNA Probe/AgNPs/PBGs

Different concentrations of TDNA were analyzed to depict the quantitative performance of the developed sensor. Different concentrations varying from 0.1 to 100 μM were employed for the hybridization of the ssDNA probe. The results concluded that TDNA is showing hybridization with the ssDNA probe, and at different concentrations a varying current response was observed, which confirmed the quantitative performance of the developed sensor. The results obtained were in line with the earlier reported sensors. Upon increasing the concentrations of TDNA, there was a decreased current response as more insulating layers of biological recognition element retarded the electron transfer. A linear relationship was found between the log value of the TDNA concentration and the anodic current value, and a good r^2^ value was found. The detection limit was found to be 0.1 μM (Figure 3).

### 3.4. Optimization of DNA Probe/AgNP/PBG Platform in Terms of Temperature and Time

Optimization of the sensor is very important for the smooth functioning of the developed sensor. The performance of the sensor depends on pH, temperature, and time. Therefore, the sensor was optimized in terms of these experimental variables so that maximum response could be achieved (Figure 4). The performance of the developed sensor DNA probe/AgNPs/PBGswas extensively studied in varying different ranges of temperature and time. The cyclic voltammogram of the DNA probe/AgNPs/PBGs was observed at different temperatures ranging from 15 to 45 °C at a scan rate of 50 mV/s. The highest response was observed at 45 °C, with its peak almost approaching the peak of the CV without TDNA; this could be due to instability of the double helical structure at high temperatures, as the high-energy TDNA would not easily bind to the PDNA. Therefore, the sensor was optimized at 25 °C. The created sensor was tuned at various times to provide the best possible response at any given moment. After 20 s, there was no significant change in anodic current value as current values of 20 s and 30 s were overlapping with each other, implying that 20 s is the optimal reaction time. An ideal incubation time of 20 s was maintained prior to the anticipated sweep runs.

### 3.5. Evaluation Parameters

The developed sensor was evaluated in terms of its reproducibility, recovery, and stability (Figure 5 and Table 1). The recovery of the developed sensor was checked by adding 4 µM to the initial concentration 1 µM. The current response of the original concentration (5 µM) was similar to the current response obtained after addition. The developed sensor was able to show good recovery. CV was taken to check the reproducibility of the modified sensor. The developed sensor was checked at five replicates’ determination. The target DNA was estimated on a single day five times and it was observed that the current remained unchanged at different determinations. The response of the sensor was also estimated after one week and the sensor response was found to be almost the same. The observations demonstrated that the sensor showed high reproducibility. The developed sensor was tested for its long-term stability. The paper electrodes were checked after 15 days. The sensor showed almost the same response as it did on day one.

### 3.6. Specificity and Reliability

This PBG selectivity was estimated via analysis with the target yellow fever virus (YFV) DNA. Reactivity was observed only in the target DNA of ZIKV. Results show that there was no major change in the current peak of the DNA probe of ZIKV before or after the addition of the target DNA for YFV, respectively; the current peak remained unchanged (Figure 6). This study revealed that the target DNA YFV had no specific binding to the specific probe DNA of ZIKV. However, in presence of the target ZIKV DNA, binding occurred between the DNA probe and the target ZIKV TDNA. The bar graph in Figure 6 represents the confirmation of the above results.

### 3.7. Analysis of Experimental Sample

The CV response of the sensor when the TDNA ZIKV was spiked into a synthetic serum sample was found to be similar to the target DNA alone (Figure 7). The developed sensor was able to detect the target DNA in synthetic serum as the current response was found to be almost the same as the target DNA alone.

## 4. Discussion

The present work aimed to develop an electrochemical paper-based device for the detection of ZIKA virus target DNA, which offered many advantages, such as low sample/reagent volume, ease of preparation, and more stable printing on paper, unlike plastic substrate, and the sensor was also applied in artificial serum samples so that sensor applicability could be checked in complex matrices. The use of a state-of-the-art portable potentiostat acts as a limiting factor for the device’s LOD, as can be seen in Table 2, but this diminishes when the fact that this can be replicated in any remote corner of the world is taken into consideration, effectively making the device truly capable of providing POC diagnosis, which is crucial considering the tropical presence of the disease in question. Still, the developed genosensor has major disadvantages in that the sensor was not able to detect the direct antigen, which can be overcome by employing aptamers a biological recognition elements. The employment of aptamers as biological or structural recognition elements can be considered as a future perspective of the developed study. The advancement in technology and the development of more sensitive potentiostat devices also open the scope of lower LODs and better detection capabilities for such devices.

## 5. Conclusions

The current research describes the synthesis of electrochemical PBGs for the identification of ZIKV TDNA. AgNPs deposited on paper-based electrodes were utilized for boosting the signal response. The sensor displayed an enhanced linear range from 0.1 to 100 µM and the lower limit of detection was 0.1 µM. The usage of paper-based electrodes also put on the benefit of being disposable and lower-cost, as well as having the capability of being fabricated in-house. All of these features together make the current effort ideal for developing point-of-care systems. With a few adjustments, this approach could develop into a technology that could be employed at the bedside of ZIKV patients.

## Figures and Tables

**Figure 1 micromachines-13-02037-f001:**
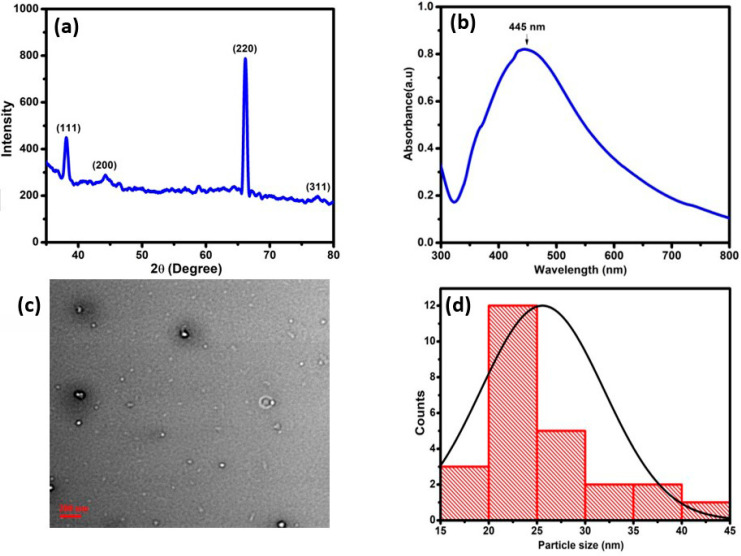
(**a**) XRD diffractogram and (**b**) UV-Vis spectroscopy; (**c**) TEM image of silver nanoparticles; (**d**) average particle size.

**Figure 2 micromachines-13-02037-f002:**
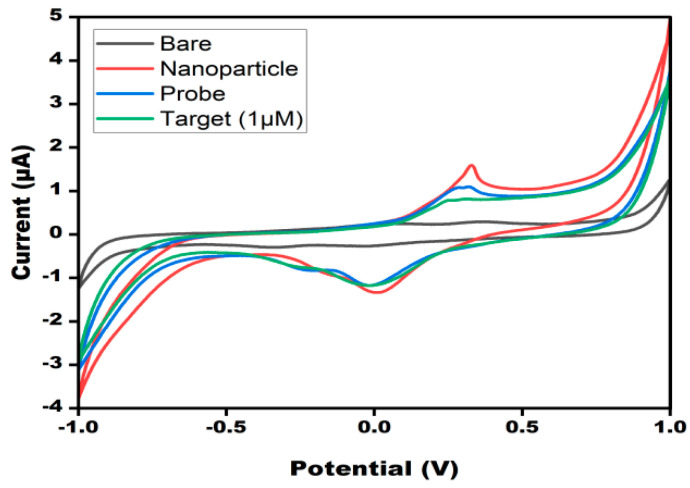
Cyclic voltammograms of 10 mM methylene blue in 0.1MKCl at bare PBGS, AgNPs/PBGS, DNA probe/AgNPs/PBGS and TDNA/PDNA/AgNPs/PBGS at 50 mV s^−1^ in the potential range from −1 V to +1 V.

**Figure 3 micromachines-13-02037-f003:**
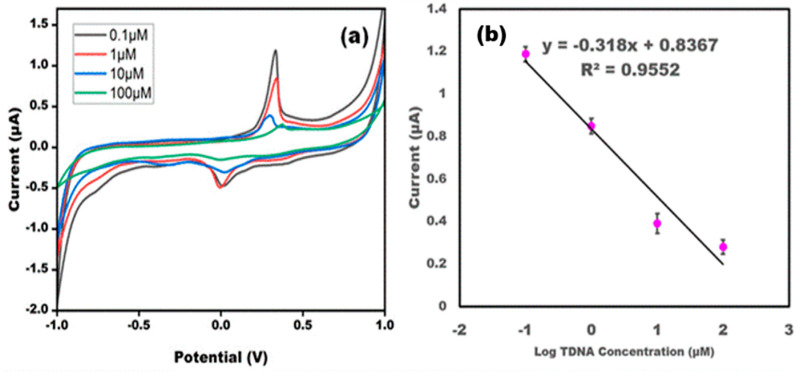
(**a**) CV of DNA probe/AgNP/PBG-modified electrode using different concentrations of ZIKV-TDNA (0.1 µM, 1 µM, 10 µM, and 100 µM) in 10 mM methylene blue in 0.1 M KCl at 50 mV s^−1^ in the potential range from −1 V to +1 V. (**b**) Linear curve of the current value and log of the target DNA concentration. The error bar represents the standard deviation of the sensor for each concentration repeated 5 times.

**Figure 4 micromachines-13-02037-f004:**
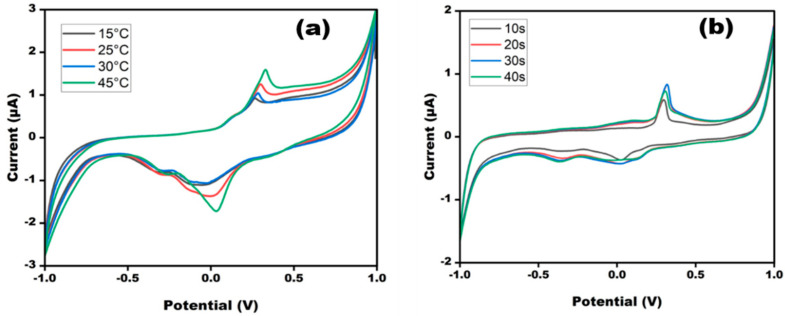
Cyclic voltammetry obtained at DNA probe/AgNPs/PBGs for different (**a**) temperature (15–45 °C) and (**b**) time (10–40 s) in 10 mM methylene blue in 0.1 M KCl at 50 mV s^−1^ in the potential range from −1 V to +1 V.

**Figure 5 micromachines-13-02037-f005:**
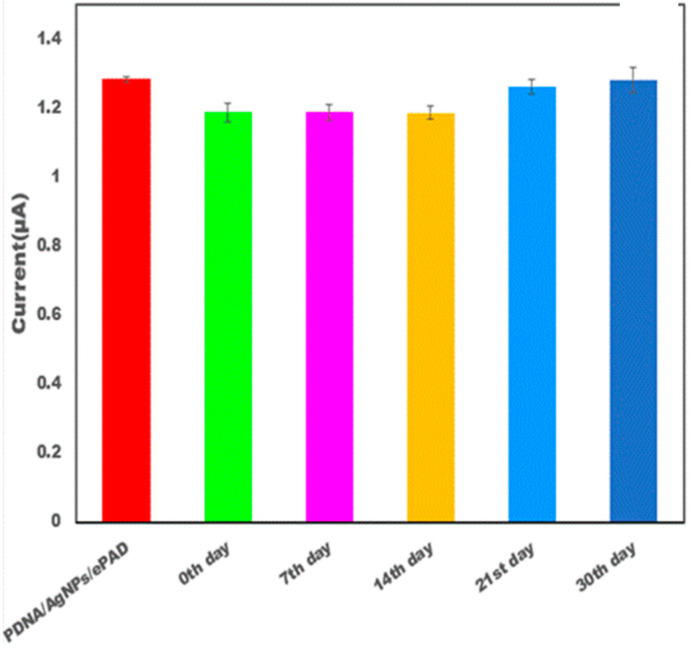
Bar graph depicting the stability of the modified electrode DNA probe/AgNPs/PBGS for TDNA on the 7th, 14th, 21st, and 30th day in 10 mM methylene blue in 0.1MKCl at 50 mV s^−1^ in the potential range from −1 V to +1 V.

**Figure 6 micromachines-13-02037-f006:**
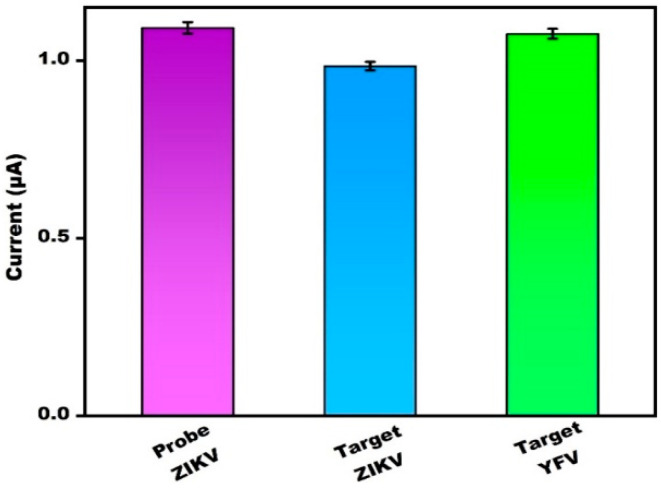
Bar graph depicting the CV confirmation with crossbar with the current value of probe ZIKV/target ZIKV and probe YFV.

**Figure 7 micromachines-13-02037-f007:**
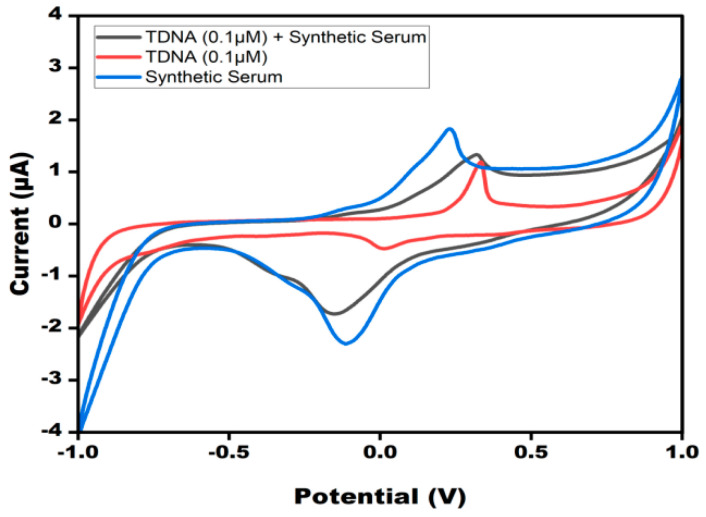
CV peak current study of TDNA of ZIKV in synthetic serum at modified DNA probe/AgNPs/PBGs in10 mM methylene blue in 0.1 M KCl at 50 mV s^−1^ in the potential range from −1 V to +1 V. CV peak current was compared with the synthetic serum, TDNA, and TDNA in synthetic serum.

**Table 1 micromachines-13-02037-t001:** Recovery test of constructed biosensor of ZIKV.

Initial Concentration (µM)	Concentration Added (µM)	Concentration Found (µM)	Recovery (Percentage)
1	4	5.1	102

**Table 2 micromachines-13-02037-t002:** Comparative study of various ZIKV-based genosensors.

Biosensors	Linear Range	LOD	References
Electrochemical genosensor for Zika virus based on a poly-(3-amino4-hydroxybenzoic acid)-modified pencil carbon graphite electrode	84.0 pM to 1.41 nM	25.4 pM	[14]
Gold nanoparticle/DNA-based nanobioconjugate for electrochemical detection of Zika virus	10 to 600 fM and from 500 fM to 10 pM of the target	0.2 and 33 fM at the SPAuE and SPCE/Au	[12]
Label-free electrochemical DNA biosensor for Zika virus identification	-	25.0 ± 1.7 nM	[15]
A sensitive label-free impedimetric DNA biosensor based on silsesquioxane-functionalized gold nanoparticles for Zika virus detection.	1.0 ×10^−12^–1.0 ×10^−6^ M	0.82 pM	[18]
Electrochemical biosensor based on surface imprinting for Zika virus detection in serum	10 fM–1 μM	9.4 fM	[19]
Acrylic-based genosensor utilizing metal salphen labeling approach for reflectometric dengue virus detection	1 × 10^−15^ M to 1 × 10^−3^ M	1.21 × 10^−16^ M	[20]
Rapid, point-of-care, paper-based plasmonic biosensor for Zika virus diagnosis	10–105 nM	1 nM	[21]
Diagnosis of Zika infection using a ZnO nanostructure-based rapid electrochemical biosensor	0.1 nM to 100 nM	1.00 pM	[22]
Cyclic voltammetric PBG-based detection of the target DNA of Zika virus	0.1 to 100 µM	0.1 µM	This present work

## Data Availability

Data will be made available on request.

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
