# Peer review of "Cyclic Voltammetric-Paper-Based Genosensor for Detection of the Target DNA of Zika Virus"

_micromachines, 2022, doi:10.3390/mi13122037_

Round 1

Reviewer 1 Report

The authors presented a study based on selective electrochemical sensor for DNA of Zika Virus detection based on Ag NPs decorated carbon-printed paper electrode. However, the results are not significant.

1.       Elaborate the novelty and research motivation of this work in the Introduction clearly, focusing on previous methods and this proposed method for the detection of Zika Virus.

2.       Check this manuscript to avoid English language errors carefully, especially grammar and spelling errors, superscripts and subscripts, etc. (e.g., AG NPs, Faria et al, NaBH4, figure 1(a))

3.       Line 183, the sensor is for detecting the cDNA of ZIKV but not ZIKV itself. Please be more accurate. Few other places have the same issue.

4.       The labels in Fig.3 and Fig.5 need to be fixed.

5.       There are a few places where the author mentioned other earlier publications without citing them.

6.       Fig. 4 mismatches to the experiment description in “Optimisation of Physico-chemical parameters”. And the observation lacks an explanation of why it happened like that.

7.       Discussion regarding Fig.5a is missing.

8.       How to confirm the detection sensitivity, selectivity, stability and repeatability of this proposed method during the practical determination process of DNA in real samples?

9.       In table 1, the detection range for this work is about 13 nM to 13µM (considering MW is 7064 DA) which is not better than other published work.

10.   The authors need to provide more explanations and discussions, do not only describe experimental results. Improve the whole manuscript.

Therefore, I cannot suggest its publication

Author Response

attached file

Reviewer 2 Report

Interesting research, however some confirmations needed to improve the quality of the manuscripts as follows:
- Abstract write as stuctured, I saw generally article in this journal was not write as stuctured abstract.

- Why do you use silver nanoparticles in this research, instead of other nanoparticles?
- CV usually is less sensitive compare to another electrochemical method. Maybe you have a reason to select the CV, instead of DPV, SWV or other method in electrochemical?
- What was the interaction between carbon working electrode and silver nanoparticle? is this physica absorption? or you depostie the nanoparticle on the working electrode by electrodeposition? and also the PDNA and Ag nanoparticle interaction, it was and physical or chemical bonding?

- Why you use MB in KCl to determine the interaction of each modification in the electrode?

- since the detection principles was binding the targetted DNA to the working electrode, which means, lowering the CV peaks, it is better the blank have as high as possible of redox peaks. You use 20 uL of silver nanoparticle on the working electrode, maybe the higher amount would be better, isn't is?

- it is should be better if the TEM image in Fig 1 has a clear scale

- Please describe the advantages of this research compare to similar principle or analytes, e.g. https://doi.org/10.1016/j.procbio.2018.08.020, https://link.springer.com/article/10.1007/s00604-020-04568-1

Author Response

attached file

Reviewer 3 Report

Title: Cyclic voltammetric-Paper-Based Genosensor for detection of 2 the target DNA of zika virus

Current form of manuscript is Revision required

1. Authors didn't explain, size of the Silver nanoparticles. Authors must explain with evidence.

2. XRD data is not clear (Figure 1a)

3. Authors must explain detection mechanishm in the sensor. Redox peak is obtained based on Figure 2 and 3. Authors must explain clearly. 

4. What is the advantages in the genosensors? authors must explain with following articles https://link.springer.com/article/10.1007/s00604-022-05463-7; https://www.sciencedirect.com/science/article/pii/S2590137022001273;

https://www.sciencedirect.com/science/article/pii/S0013468621007799

5. There are many grammatical and typographical errors. Please check the manuscript and refine carefully.

Author Response

attached file

Round 2

Reviewer 1 Report

The revised manuscript is well-written. However, I cannot recommend publishing this article.

The only novelty of this work is having ZIKV TDNA detection on "a low-cost, simple, portable and reliable device", however, the sensor has a significantly degraded detection limit and detection range with a well-known sensing method.

Author Response

attached file

Reviewer 2 Report

You have addressed all my comments and suggestions

Author Response

attached file
